# Evaluation of forearm vascular resistance during orthostatic stress: Velocity is proportional to flow and size doesn't matter

V. E. Claydon[1]*, J. P. Moore[2], E. R. Greene[3], O. Appenzeller[4], R. Hainsworth[5]

1 Department of Biomedical Physiology and Kinesiology, Simon Fraser University, Burnaby, British Columbia, Canada, 2 School of Sport, Health & Exercise Sciences, Bangor University, Bangor, Gwynedd, United Kingdom, 3 Department of Biology and Chemistry, New Mexico Highlands University, Las Vegas, New Mexico, United States of America, 4 Department of Neurology, New Mexico Health Enhancement and Marathon Clinics Research Foundation, Albuquerque, New Mexico, United States of America, 5 Division of Cardiovascular and Neuronal Remodeling, Faculty of Medicine, University of Leeds, Leeds, United Kingdom

* victoria_claydon@sfu.ca

## Abstract

### Background

The upright posture imposes a significant challenge to blood pressure regulation that is compensated through baroreflex-mediated increases in heart rate and vascular resistance. Orthostatic cardiac responses are easily inferred from heart rate, but vascular resistance responses are harder to elucidate. One approach is to determine vascular resistance as arterial pressure/blood flow, where blood flow is inferred from ultrasound-based measurements of brachial blood velocity. This relies on the as yet unvalidated assumption that brachial artery diameter does not change during orthostatic stress, and so velocity is proportional to flow. It is also unknown whether the orthostatic vascular resistance response is related to initial blood vessel diameter.

### Methods

We determined beat-to-beat heart rate (ECG), blood pressure (Portapres) and vascular resistance (Doppler ultrasound) during a combined orthostatic stress test (head-upright tilting and lower body negative pressure) continued until presyncope. Participants were 16 men (aged 38.4±2.3 years) who lived permanently at high altitude (4450m).

### Results

The supine brachial diameter ranged from 2.9–5.6mm. Brachial diameter did not change during orthostatic stress (supine: 4.19±0.2mm; tilt: 4.20±0.2mm; -20mmHg lower body negative pressure: 4.19±0.2mm, p = 0.811). There was no significant correlation between supine brachial artery diameter and the maximum vascular resistance response (r = 0.323; p = 0.29). Forearm vascular resistance responses evaluated using brachial arterial flow and velocity were strongly correlated (r = 0.989, p<0.00001) and demonstrated high equivalency with minimal bias (-6.34±24.4%).

**Data Availability Statement:** Due to legal and ethical restriction, data cannot be made publicly available. Data will be made available upon request; however, only aggregated data may be in the public

domain according to the stipulations from our research ethics board with respect to the maintenance of confidentiality. Additional published or public analyses would only be permitted with ethics approval for secondary data access, and only with aggregated analyses. Requests can be sent to Dina Shafey, Associate Director, Office of Research Ethics, Simon Fraser University (dshafey@sfu.ca).

**Funding:** This work was supported by funding from the New Mexico Health Enhancement and Marathon Clinics Research Foundation (OA), The Physiological Society (VEC) and the University of Leeds (RH). The funders had no role in study design, data collection and analysis, decision to publish, or preparation of the manuscript.

**Competing interests:** The authors have declared that no competing interests exist.

## Discussion

During severe orthostatic stress the diameter of the brachial artery remains constant, supporting use of brachial velocity for accurate continuous non-invasive orthostatic vascular resistance responses. The magnitude of the orthostatic forearm vascular resistance response was unrelated to the baseline brachial arterial diameter, suggesting that upstream vessel size does not matter in the ability to mount a vasoconstrictor response to orthostasis.

## Introduction

The assumption of an upright posture imposes a significant challenge to the regulation of blood pressure [1]. The ability to appropriately compensate for orthostatic fluid shifts through baroreflex-mediated increases in heart rate and vascular resistance is a key determinant of orthostatic tolerance, defined as the ability to maintain haemodynamic stability when upright, and therefore to tolerate standing [1,2]. Impaired orthostatic tolerance is associated with fainting, or syncope. The relative contributions of cardiac and vascular responses to orthostatic cardiovascular control have been the subject of much debate [3]. This may partly reflect that orthostatic cardiac responses are easy to infer from changes in heart rate, but sympathetically-mediated vascular resistance responses are technically more challenging to elucidate. Common techniques for evaluation of vascular responses include measurements of efferent muscle sympathetic nerve activity [4] or inference based on changes in circulating catecholamine levels or blood pressure [5]. However, changes in blood pressure alone cannot be attributed to vascular resistance responses because they fail to account for alterations in blood flow. Muscle sympathetic nerve activity or catecholamine release represent excellent tools for determining alterations in sympathetic outflow or neurotransmitter release, but do not capture the effector organ response, which may be affected by changes in vascular transduction [6].

One approach for the determination of vascular resistance is to calculate it using the haemodynamic equivalent of Ohm's law, where resistance is equal to mean arterial pressure divided by blood flow. Often the blood flow is inferred from Doppler ultrasound measurements of changes in blood velocity (in cases where continuous measures of arterial diameter are not available), typically based on recordings from the brachial artery, with the presumption that velocity will be proportional to flow as long as the diameter of this conduit vessel remains constant, with the site for regulation of vascular resistance being the downstream arterioles [7–9]. This approach is beneficial because it measures the end-organ response, and can provide continuous and non-invasive estimates of vascular resistance. The caveat to this approach is that the diameter of the brachial artery has never been reported during severe orthostatic stress, and if it does change during profound sympathetic activation, could represent a significant confound in the interpretation of vascular resistance responses. It is also not known whether the ability to mount an orthostatic vascular resistance response is related to the initial blood vessel diameter. Accordingly, we aimed to: (i) measure brachial artery diameter during maximal sympathetic activation with an orthostatic stress test continued until presyncope; (ii) determine whether maximal forearm vascular resistance responses to orthostatic stress are related to baseline brachial arterial diameter; (iii) examine the equivalency of responses of forearm vascular resistance determined as mean arterial pressure divided by brachial blood flow, and using brachial artery velocity as a proxy for flow. We hypothesised that the diameter of this conduit arterial vessel would remain constant during orthostatic stress, and therefore that measures of vascular resistance determined using flow, or resistance as a proxy for flow, would

be equivalent. We hypothesised that the vascular resistance response evoked would be independent of the baseline blood vessel diameter.

## Methods

### Ethical approval

The study was approved by the Institutional Review Board of the Ladakh Institute of Prevention (for the study of environmental, occupation, lifestyle related and high altitude diseases) and was conducted in accordance with the Declaration of Helsinki (2002) of the World Medical Association. All participants provided written informed consent in their native language, with translators present throughout testing to explain procedures and answer any questions raised by the participants.

### Participants

These data were collected as one part of a larger one-month long field study conducted in 2007 examining cardiovascular control in residents of high altitude. While the present analyses do not address the primary research question of this field study, these data provide a fortuitous opportunity to consider the validity of the assumptions that accompany the use of Doppler measurements of blood velocity in determining vascular resistance responses, and are not likely to be influenced by the altitude at which the tests were performed. Participants were 16 men who were born and lived permanently at high altitude (Korzok, 4450m) in the Ladakh region of the Himalayan mountain range. Testing was conducted in a nearby field research site at 4606m elevation. Their mean age was 38.4±2.3 years, height 1.60±0.1m, and weight 56.8 ±1.7kg. None of the participants was taking any medications. Participants were excluded if they had significant acute or chronic clinical conditions, or had visited altitudes below 2000m in the 2 months prior to the study. Studies were conducted in an environment that was thermoneutral for our clothed participants.

### Orthostatic stress testing

Orthostatic stress was applied using graded head-upright tilting combined with lower body negative pressure as described previously [7,10]. In brief, participants rested in the supine position for 20 minutes to allow stabilisation of cardiovascular haemodynamics. They were then head-up tilted to 60˚ for a further 20 minutes. After this time, while still tilted, graded lower body negative pressure was applied at -20mmHg, -40mmHg, and -60mmHg for ten minutes at each level. The test was terminated if the participant completed the entire protocol, if they requested to stop, of if they experienced symptoms or signs of presyncope associated with hypotension (systolic blood pressure <80mmHg).

Throughout testing we continuously recorded heart rate and rhythm (lead II electrocardiograph, Hewlett Packard, 78352C) and non-invasive beat-to-beat blood pressure using finger plethysmography (Portapres Model 2, TNO-TPD Biomedical Instrumentation) from the middle finger while supported at heart level. Mean arterial pressure was determined (diastolic arterial pressure + 1/3 (pulse pressure)) from a 15-second average taken every two minutes during the test.

We also determined brachial artery blood velocity and diameter using image-guided Doppler flowmetry (HP Sonos 100). A 7.5MHz linear array transducer was positioned overlying the brachial artery and held in place with a constant angle of insonation. Time averaged internal lumen diameter (end diastolic diameter + 1/3 (peak systolic diameter–end diastolic diameter) was determined using image-guided M-mode waveforms by blinded assessment with a spatial resolution of ~0.2mm and inter-operator variability of ±5% (95% confidence limit). The

brachial artery diameter was measured in duplicate, over ten heart beats, at three time points during the test: the final minute of the supine rest period; the final minute of head-upright tilting; and the final minute of the -20mmHg lower body negative pressure phase. Measures of brachial diameter were not possible at the end of the -40mmHg lower body negative pressure phase because few participants tolerated this phase, and those that did were imminently pre-syncopal–our focus at that time was prompt termination of the test to prevent frank syncope in our participants. Measurements of mean brachial artery blood velocity (area under the curve) were determined from a 15-second average every two minutes throughout the test (over the same beats for which the blood pressure was determined).

Forearm vascular resistance was calculated as mean arterial pressure / brachial blood flow ($FVR_{flow}$) or velocity ($FVR_{velocity}$). Resistance responses (both determined as $FVR_{flow}$ and $FVR_{velocity}$) during orthostatic stress were expressed as percentage changes from the supine baseline value at each time point and indicated as % $FVR_{flow}$ or %$FVR_{velocity}$ respectively. The maximum vascular resistance response during the upright portion of the test was determined from the peak value of the continuous measures of %$FVR_{velocity}$ and taken as the ability to mount baroreflex-mediated vasoconstriction in response to the orthostatic stimulus.

## Statistical analyses

All data are expressed as mean±standard error. Data were tested for normality using the Kolmogorov-Smirnov assumption. Comparisons of vascular parameters at the different stages of the test were performed using one-way repeated measures ANOVA. Correlations were examined using the Pearson or Spearman correlation coefficient for parametric and non-parametric data respectively. Equivalency between measures was determined using Bland-Altman analyses. Statistical significance was assumed at the level of $P<0.05$.

## Results

Of the 16 participants, 15 experienced presyncope during the orthostatic stress test, necessitating termination of the test. In all cases this occurred at high levels of orthostatic stress, during either the -40mmHg (n = 8) or -60mmHg lower body negative pressure phase (n = 6). The remaining participant completed the entire procedure and their test was terminated at the end of the final phase (-60mmHg of lower body negative pressure).

All participants exhibited vasoconstriction in response to the test, with a maximum vascular resistance (%$FVR_{velocity}$) response of +244.1±36.3% occurring after 31±2.8 minutes of orthostatic stress (corresponding to the first minute of the -40mmHg lower body negative pressure phase) (Table 1).

The supine brachial diameter ranged from 2.9–5.6mm between participants. Within participants, brachial diameter did not change during the orthostatic stress (supine: 4.19±0.2mm; tilt: 4.20±0.2mm; -20mmHg lower body negative pressure: 4.19±0.2mm, p = 0.811) (Fig 1A).

We considered whether the baseline diameter influenced the ability to mount a vascular response to the orthostatic stress; there was no significant correlation between the supine brachial artery diameter and the maximum vascular resistance response (Fig 1B).

We performed correlations between assessments of brachial velocity and flow over the time periods at which they were simultaneously acquired (Fig 2). There were strongly significant correlations between the variables for all data combined (r = 0.917, p<0.00001) as well as within each phase of the test considered independently (supine: r = 0.903, p<0.00001; tilt: r = 0.930, p<0.00001; -20mmHg lower body negative pressure: r = 0.835, p<0.00001).

FVR responses to imposed stimuli such as orthostatic stress are normally expressed as the percentage change from a resting value, to normalize for baseline differences between

**Table 1. Forearm vascular parameters.**

| | MAP mmHg | Diameter cm | Velocity cm.s$^{-1}$ | FVR$_{velocity}$ mmHg/cm.s$^{-1}$ | %FVR$_{velocity}$ % | Flow cm$^3$.s$^{-1}$ | FVR$_{flow}$ mmHg/cm$^3$.s$^{-1}$ | %FVR$_{flow}$ % |
|---|---|---|---|---|---|---|---|---|
| Supine | 78.9±2.9* | 4.19±0.2 | 4.52±0.7 | 25.0±3.1*‡ | - | 59.1±12.6 | 2.39±0.5 | - |
| Tilt | 83.4±3.2* | 4.20±0.2 | 3.78±0.6*‡ | 36.2±5.6*‡ | 57.8±21.7*‡ | 45.0±8.6 | 3.54±0.8 | 61.8±23.1 |
| LBNP$_{20}$ | 79.5±4.9* | 4.19±0.2 | 2.49±0.3† | 42.7±5.1*‡ | 101.7±28.1*‡ | 32.6±6.0† | 3.81±0.8† | 104.8±33.4 |
| LBNP$_{40}$ | 78.6±4.9* | - | 2.59±0.7† | 49.2±12.4 | 96.0±43.5*‡ | - | - | - |
| Max FVR (%) | 78.9±4.7* | - | 1.52±0.2† | 66.4±8.0 | 244.1±36.3 | - | | - |
| Presyncope | 54.3±5.7 | - | 1.38±0.1† | 61.8±11.5 | 164.2±44.2 | - | | - |

MAP was significantly lower during presyncope compared to the other test time points. There were no differences in brachial arterial diameter between test phases. In general, measures of velocity and flow decreased, while measures of FVR increased, during the orthostatic stress. Max FVR (%) reflects the maximum percentage change in FVR$_{velocity}$ during the test. There were no significant differences in the percentage increase in brachial artery FVR at comparable time points relative to supine between measures determined using velocity (%FVR$_{velocity}$) or measures determined using flow (%FVR$_{flow}$).

* denotes significant difference compared to corresponding values at presyncope (p<0.05)

† denotes significant difference compared to corresponding values during supine

‡ denotes significant difference from corresponding values at the time of the maximal FVR response. Abbreviations: FVR, forearm vascular resistance; FVR$_{flow}$, FVR determined as mean arterial pressure divided by brachial arterial flow; FVR$_{velocity}$, FVR determined as mean arterial pressure divided by brachial arterial velocity; %FVR$_{velocity}$, the percentage change in FVR$_{velocity}$ relative to supine; %FVR$_{flow}$, the percentage change in FVR$_{flow}$ relative to supine; LBNP$_{20}$, -20mmHg lower body negative pressure combined with head-upright tilt; LBNP$_{40}$, -40mmHg lower body negative pressure combined with head-upright tilt; MAP, mean arterial pressure; Supine, at the end of the supine period; Tilt, after 20 minutes of 60˚ head-upright tilting.

individuals. We compared the percentage change in FVR determined using brachial arterial blood flow (%FVR$_{flow}$), and brachial arterial velocity as a proxy for flow (%FVR$_{velocity}$), at the end of the 20 minutes of head-upright tilting (Tilt), and after a further 10 minutes of head-upright tilting combined with -20mmHg lower body negative pressure (LBNP$_{20}$) (**Fig 3**). We found strong and significant correlations between %FVR$_{flow}$ and %FVR$_{velocity}$ (r = 0.989, p<0.00001) for the data combined, and when considering tilt (r = 0.988, p<0.00001) and LBNP$_{20}$ (r = 0.911, p<0.00001) separately. Bland-Altman analyses showed high agreement with minimal bias (-6.34%) between measures of %FVR$_{flow}$ and %FVR$_{velocity}$. There were no significant differences in %FVR$_{flow}$ and %FVR$_{velocity}$ determined at the end of 20 minutes of 60˚ head-upright tilt, or after a further 10 minutes of 60˚ head-upright tilt combined with LBNP.

## Discussion

### Impact of orthostatic stress on brachial artery diameter

We have demonstrated that the diameter of the brachial artery does not change leading up to maximal increases in sympathetic drive to the peripheral resistance vessels induced during orthostatic stress. This is important because forearm blood velocity is often used as a proxy for forearm blood flow, based on the assumption that velocity will be proportional to flow providing the diameter of the insonated vessel does not change. Our data support this notion. Indeed, measures of brachial velocity and flow were strongly correlated in general, and within each phase of the test. Measures of forearm vascular resistance responses (normalised to the baseline level) whether derived using brachial arterial flow or velocity were similar and strongly correlated during orthostatic stress, demonstrating strong agreement with minimal bias. Accordingly, the consideration of peripheral vascular resistance responses based on measures of brachial artery blood velocity represents a valid approach for the continuous non-invasive quantification of human vascular resistance responses in cases where continuous measures of

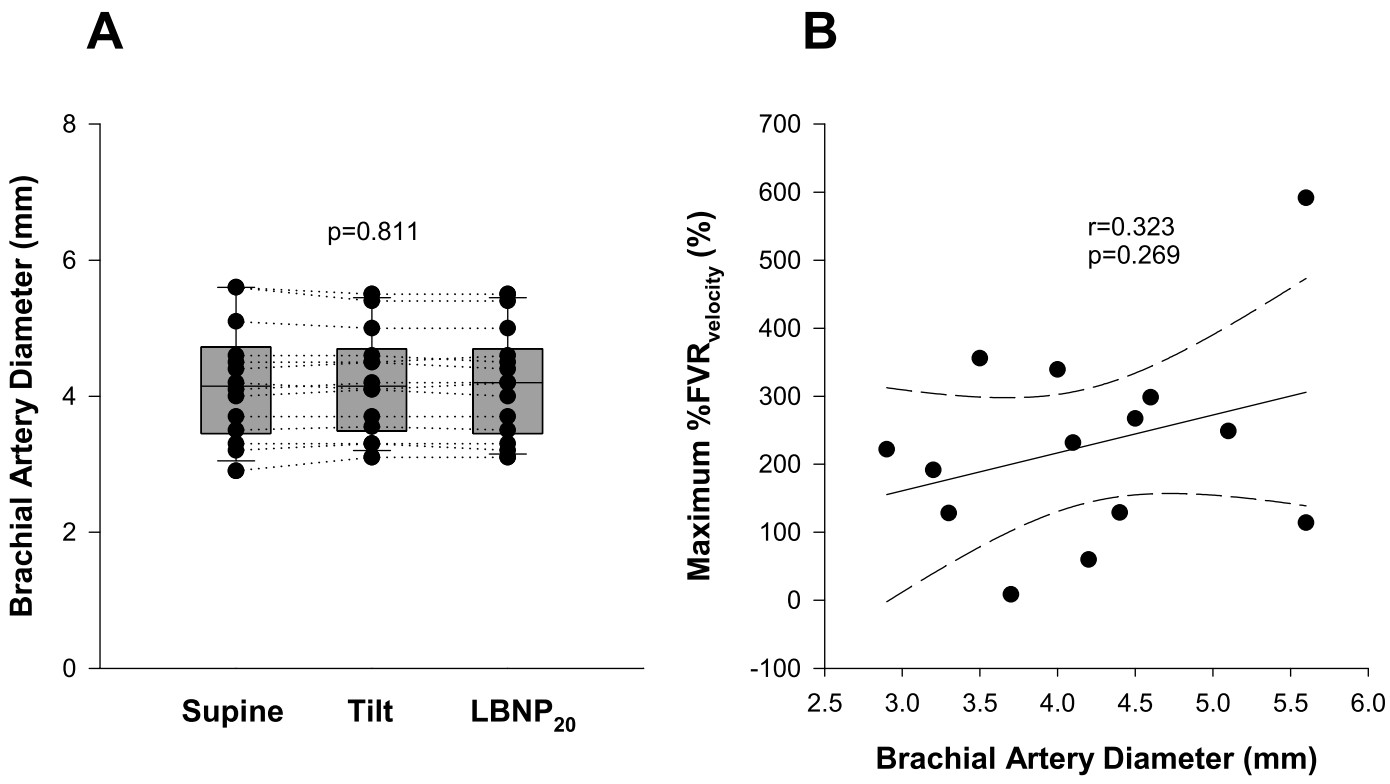

**Fig 1. Influence of orthostatic stress on brachial artery diameter and forearm vascular resistance responses (FVR).** (A) There were no significant differences in brachial artery diameter between measurements at the end of the supine period (Supine), after 20 minutes of 60° head-upright tilt (Tilt) or after a further 10 minutes of head-upright tilt with combined -20mmHg lower body negative pressure (LBNP$_{20}$). (B) There was no significant relationship between the supine brachial artery diameter and the maximum vascular resistance response to orthostatic stress expressed as the percentage change in FVR$_{velocity}$ relative to supine.

brachial arterial diameter (and therefore brachial blood flow) are not available, even during profound sympathetic activation elicited by severe orthostatic stress.

These data are in keeping with a previous report that showed no change in brachial diameter from baseline (3.9±0.2mm) to maximal forearm exercise (3.9±0.1mm) [11]. Brachial artery diameter is also reported to be unaffected when comparing the supine, seated and active standing posture [12], in response to simulated orthostatic stress using supine lower body negative pressure [13–16], and in response to changes in limb transmural pressure [17]. Increases in forearm blood flow during thermoregulatory stimuli (increase in skin temperature to 38°C) were also unaccompanied by significant changes in brachial artery diameter [18].

The impact of manipulation of end-tidal gases on brachial artery diameter is less clear, with profound increases in brachial blood flow during hypercapnia, but ambiguity as to whether this primarily occurs through dilatation of the brachial artery, or downstream arterioles [19,20]. Isocapnic hypoxia with an end tidal partial pressure of oxygen (P$_{ET}$O$_2$) 50mmHg produced a small but significant increase in brachial artery diameter +0.2±0.2mm (p = 0.01) [21]. Less severe hypoxia (P$_{ET}$O$_2$ 75mmHg) did not influence the diameter [21]. These data suggest that while there seems to be negligible impact of sympathetic stimulation during orthostatic stress on brachial artery diameter, vascular resistance responses based on brachial arterial velocity measurements collected during conditions of profound hypoxia or hypercapnia should be treated with more caution.

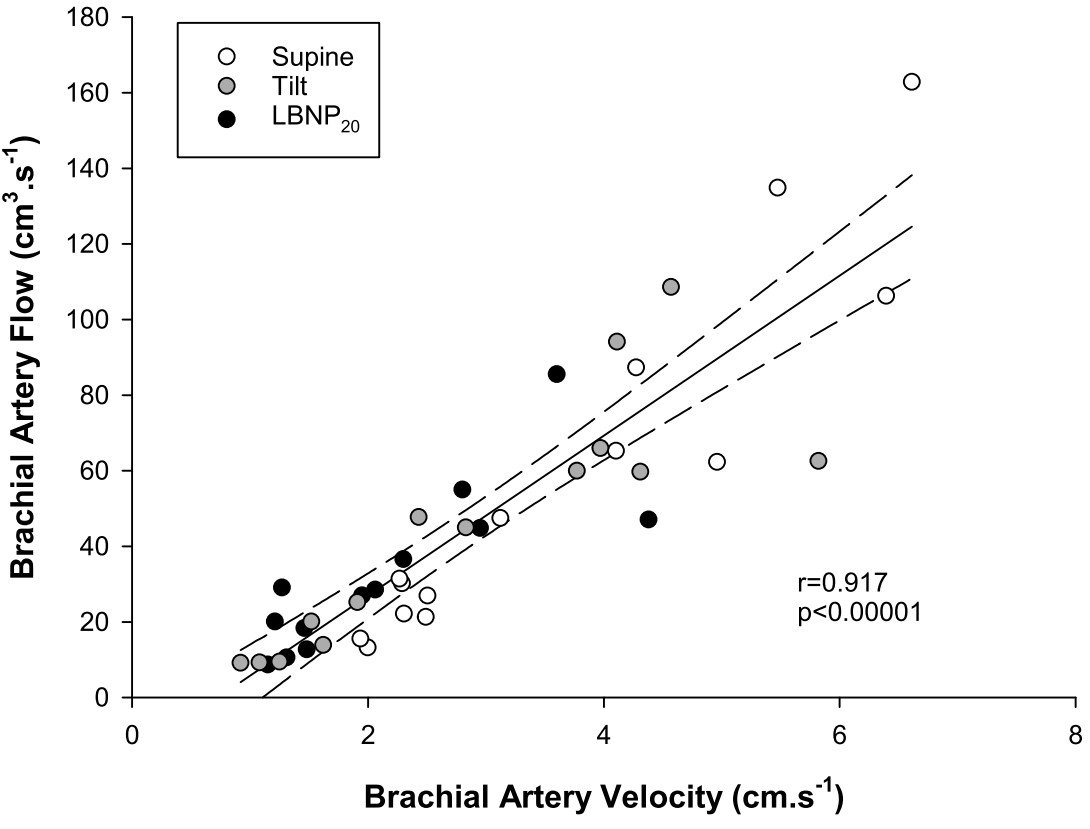

**Fig 2. Relationship between simultaneous measurements of brachial artery velocity and flow at baseline and during orthostatic stress.** There was a strong correlation between measurements of brachial artery velocity and flow at the end of the supine period (Supine, white), head-upright tilt (Tilt, grey) and head-upright tilt with combined -20mmHg lower body negative pressure (LBNP$_{20}$, black).

## Impact of blood vessel diameter on vasoconstrictor reserve

We also showed that the forearm vascular resistance response during maximal orthostatic stress (confirmed by the development of presyncope in all but one of the participants) was unrelated to the baseline brachial artery diameter. This implies that in individuals with blood vessels of different sizes, the ability to mount a vascular response is independent of the size of the upstream blood vessel, at least in males in a thermoneutral environment.

This might appear to be at odds with the notion of vasoconstrictor reserve, whereby the ability to mount a sympathetic vascular response is an important determinant of orthostatic tolerance [4,22–24]. This response might be expected to be impaired if the vessel is already in a preconstricted state at baseline, with a presumed reduction in vasoconstrictor reserve.

One possible reason for the disconnect between forearm vascular resistance responses and resting brachial arterial diameter may be that the response of vascular resistance measured in the forearm is not mediated by the brachial artery but rather by smaller downstream arterioles (as evidenced by the lack of change in brachial artery diameter during orthostatic stress demonstrated in the present study) and so is unaffected by the initial diameter of this vessel.

In addition, this uncoupling of response capability from initial vessel diameter is exemplified by the largest vascular resistance responses being initiated by the small arterioles, not the

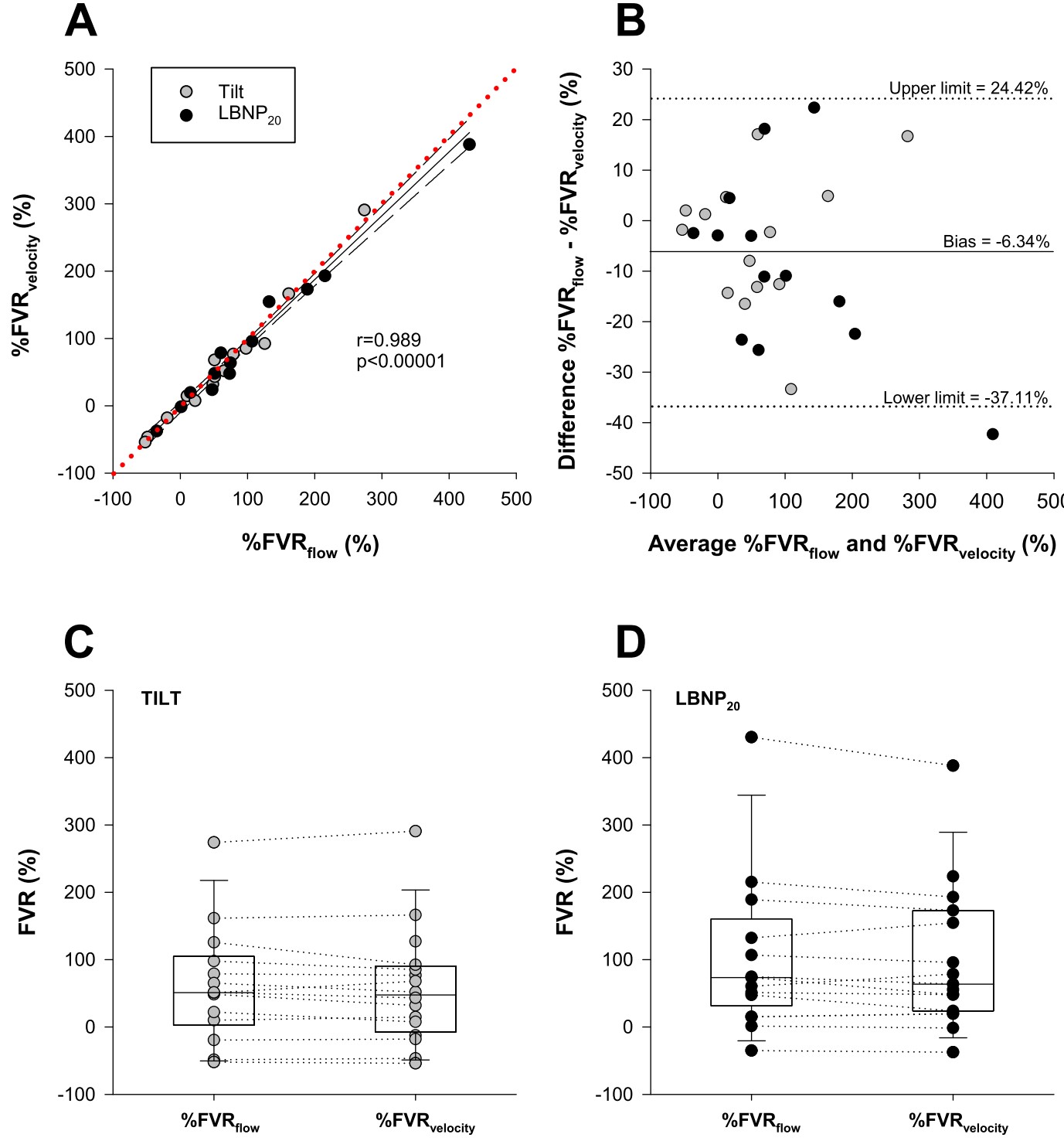

**Fig 3. Equivalency between FVR responses to orthostatic stress (percentage change from supine) determined using mean arterial pressure divided by brachial arterial flow (%FVR_flow) or using velocity as a proxy for flow (%FVR_velocity).** (A) There was a strong correlation between measurements of %FVR_flow and %FVR_velocity during orthostatic stress (Tilt, grey) and head-upright tilt with combined -20mmHg lower body negative pressure (LBNP_20, black). Red dotted line denotes the line of identity. (B) Bland-Altman analyses showed high agreement with minimal bias between measures of %FVR_flow and %FVR_velocity. There were no significant differences in %FVR_flow and %FVR_velocity determined at the end of 20 minutes of 60° head-upright tilt (C) or after a further 10 minutes of 60° head-upright tilt combined with LBNP (D). Abbreviations: FVR, forearm vascular resistance.

larger conduit arteries, and the observation that heat-stress induced vasodilatation impairs orthostatic tolerance [25,26] but does not affect the capacity of the vascular baroreflex response [27]–it is merely offset by the associated thermal vasodilatation.

## Limitations

The primary limitation of this study is that many of the participants had rather large vascular resistance responses to orthostatic stress. This may have impacted the ability to detect a relationship between brachial artery diameter and the maximum vascular resistance response, because there were few individuals with responses at the lower end of the spectrum.

An additional limitation is that the participants in this study were all permanent residents at high altitude, and the associated hypoxia and hypocapnia may have influenced their cardiovascular control. However, we consider it unlikely that the *site* for regulation of vascular resistance in these individuals (downstream arterioles as opposed to the conduit arteries) would be different from sea-level dwellers, even if the magnitude of the response were affected. Indeed, the diameters we recorded were similar to those reported in a previous report in altitude residents that also found no difference in brachial artery diameter between altitude dwellers and lowland residents [28].

Our measurements were made in males–we cannot be certain that these findings would extend to females. However, while women generally mount smaller vascular resistance responses to orthostatic stress than males, it is unlikely that the site for regulation of resistance would change between the sexes.

Finally, while we measured brachial arterial diameter during severe orthostatic stress, we did not take measurements at the moment of presyncope (because the emphasis at this time was on the rapid termination of the test and resolution of the presyncopal event). However, it should be noted that the measures during lower body negative pressure were taken within one minute of the timing of the maximum vascular resistance response of the cohort and close to presyncope for the majority of individuals. Nevertheless, we cannot exclude the possibility that there are changes in brachial arterial diameter at the moment of presyncope that would confound the use of brachial velocity as a proxy for changes in flow.

## Conclusions

We showed that during severe orthostatic stress continued until presyncope the diameter of the brachial artery does not change, and therefore forearm blood velocity measured using brachial ultrasound is proportional to forearm blood flow. This observation supports the use of Doppler-based measurements of brachial blood velocity for the accurate determination of continuous non-invasive vascular resistance responses during orthostatic stress in cases where the vessel diameter is not known. We also showed that the magnitude of the orthostatic forearm vascular resistance response was unrelated to the baseline brachial artery diameter, suggesting that size does not matter in the context of using brachial arterial measurements as a proxy for an individual's ability to mount a vasoconstrictor response to orthostasis.

## Author Contributions

**Conceptualization:** V. E. Claydon, E. R. Greene, R. Hainsworth.

**Data curation:** V. E. Claydon.

**Formal analysis:** V. E. Claydon, E. R. Greene, R. Hainsworth.

**Funding acquisition:** V. E. Claydon, O. Appenzeller, R. Hainsworth.

**Investigation:** V. E. Claydon, E. R. Greene, O. Appenzeller, R. Hainsworth.

**Methodology:** V. E. Claydon, E. R. Greene, O. Appenzeller, R. Hainsworth.

**Project administration:** O. Appenzeller, R. Hainsworth.

**Resources:** R. Hainsworth.

**Visualization:** V. E. Claydon.

**Writing – original draft:** V. E. Claydon.

**Writing – review & editing:** V. E. Claydon, J. P. Moore, E. R. Greene, O. Appenzeller, R. Hainsworth.

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
