## [Decision Letter · Decision Letter 0]

26 Sep 2019

PONE-D-19-14796

Evaluation of forearm vascular resistance during orthostatic stress: velocity is proportional to flow and size doesn’t matter

PLOS ONE

Dear Dr Claydon,

Thank you for submitting your manuscript to PLOS ONE. After careful consideration, we feel that it has merit but does not fully meet PLOS ONE’s publication criteria as it currently stands. Therefore, we invite you to submit a revised version of the manuscript that addresses the points raised during the review process.

There are a some methodological queries to be addressed and issues requiring clarification before this can be considered. 

We would appreciate receiving your revised manuscript by Nov 10 2019 11:59PM. To enhance the reproducibility of your results, we recommend that if applicable you deposit your laboratory protocols in protocols.io, where a protocol can be assigned its own identifier (DOI) such that it can be cited independently in the future. For instructions see: http://journals.plos.org/plosone/s/submission-guidelines#loc-laboratory-protocols

We look forward to receiving your revised manuscript.

Kind regards,

Christopher Torrens

Academic Editor

PLOS ONE

Journal Requirements:

2. Please provide more information regarding the setting (e.g. locations, relevant dates, periods of recruitment, data collection) and the selection criteria of participants. Please also provide more information regarding participants' baseline characteristics.

Additional Editor Comments (if provided):

This was an interesting paper but it could benefit from some clearer presentation as it was difficult to follow at times.

In terms of the test termination, are you saying the only one participants made it through to the -80 mmHg and that the rest terminated from 31 mins (1 min at -40 mmHg) onward? If so it would be nice to see the data of time to termination if only to make things clearer.

Reviewers' comments:

Reviewer's Responses to Questions

**Comments to the Author**

1. Is the manuscript technically sound, and do the data support the conclusions?

Reviewer #1: Partly

Reviewer #2: Yes

2. Has the statistical analysis been performed appropriately and rigorously? 

Reviewer #1: Yes

Reviewer #2: I Don't Know

3. Have the authors made all data underlying the findings in their manuscript fully available?

Reviewer #1: Yes

Reviewer #2: Yes

4. Is the manuscript presented in an intelligible fashion and written in standard English?

Reviewer #1: Yes

Reviewer #2: Yes

5. Review Comments to the Author

Reviewer #1: Thank you for your contribution, as this is a well written manuscript with an interesting addition to the study of vascular resistance and orthostatic hypotension. While I find this manuscript interesting and well written, I have questions regarding some of the content.

The first question regarding the methodology is the measurement of the brachial artery and not an artery closer to the lower extremities. The location of the brachial arteries makes them less likely to react to the changes in flow, as the increase in flow is going to be in the lower extremities during orthostatic hypotension. The logical argument for using the brachial artery is that brachial artery measurement and flow-mediated dilation measurements are well validated. However, in your discussion (line 207-211) you cite a study that combats the validity of FMD measurements in youth and adults. There are hundreds of studies validating the use of FMD for a variety of populations, so I'm lost as to why you would include this study to support your findings. Citing Robergs (1997) brings question as to why you would ever use ultrasound as a tool for measuring vessel diameter changes in response to flow alterations.

In line 126 the authors state the occurrence of vasoconstriction as evidenced by the increase in vascular resistance. With no change in brachial diameter, I recommend the authors be more specific regarding the location of vasoconstriction.

Thank you again for your contribution, and I look forward to seeing the edited version of this manuscript.

Reviewer #2: Claydon et. al. compared the forearm vascular resistance-- estimated using both brachial arterial flow and velocity.-- in response to supine, tilt, with and without different levels of simulated hypotension (LBNP). Strong correlations were found using both methods. The diameter of the brachial artery remains constant during orthostatic stress and no correlation was found between the basal (supine) diameter and the maximal vascular resistance response. The authors conclude that Doppler-based measurement of brachial velocity can be used to estimate vascular resistance when vascular diameters are unknown. The vascular size does not matter in the context of an individual’s ability to mount a vasoconstrictor response to orthostasis. There are several major concerns:

The current paper seems to validate a basic physiological concept that is already known and accepted--- small arteries and arterioles instead of large vessels (such as the brachial artery with diameter 3-5 mm) are the major components of resistance. Indeed, no diameter changes in the brachial artery were found during pressure compensation. Without significant diameter changes, blood velocity surely correlates with blood flow. The significance and novelty needs to be improved.

The description of data is very confusing. For example, it is stated in the Methods that in all cases presyncope occurred at high levels of orthostatic stress, during either -40mmHg or -60mmHg lower body negative pressure , and the maximum vascular resistance response occurred after 31±2.8 minutes of orthostatic stress (corresponding to the first minute of the -40mmHg lower body negative pressure phase). However, only LBNP within 20 mmHg was shown in the Results. In the limitation part, the authors mentioned that all the measures during lower body negative pressure were taken within one minute of the mean orthostatic tolerance of the cohort –close to presyncope for the majority of the cohort. Then why were data of LBNP at 40 mmHg not shown?

How was the maximal FVR% collected? The only information that I found was in the Table legend: “Max FVR (%) reflects analyses over the time point at which the maximum FVR velocity response occurred.” The description of this procedure should be explained in the Method section. In the discussion part (line 228), the authors also claim that the forearm vascular resistance response during maximal orthostatic stress is confirmed by the development of presyncope in all but one of the participants. Do these statements suggest that FVR reached maximal levels upon presyncope? If so why were diameters at Max FVR and presyncope not presented?

Although the authors pointed out as a limitation, it was still not clear how the high altitude situation may affect tolerance and vascular response to LBNP. The artery size may be unaffected but oxygen carrying capability and tissue oxygenation are likely changed in the participants.

The conclusion of “the vascular size does not matter in the context of an individual’s ability to mount a vasoconstrictor response to orthostasis” may be misleading as the vessels used for size measure were not the vessels that constricted due to increase FRV.

In Table 1, please clarify FVR calculated by velocity and VVR calculated by flow. In Figure 2, LBNP (dark close circle) data are missing. Line 167, "normalise" should be "normalize"

6. PLOS authors have the option to publish the peer review history of their article (what does this mean?). If published, this will include your full peer review and any attached files.

Reviewer #1: No

Reviewer #2: No

---

## [Author Response · Author response to Decision Letter 0]

27 Sep 2019

See uploaded response to reviewers.

---

## [Decision Letter · Decision Letter 1]

24 Oct 2019

Evaluation of forearm vascular resistance during orthostatic stress: velocity is proportional to flow and size doesn’t matter

PONE-D-19-14796R1

Dear Dr. Claydon,

We are pleased to inform you that your manuscript has been judged scientifically suitable for publication and will be formally accepted for publication once it complies with all outstanding technical requirements.

With kind regards,

Christopher Torrens

Academic Editor

PLOS ONE

Additional Editor Comments (optional):

Reviewers' comments:

Reviewer's Responses to Questions

**Comments to the Author**

1. If the authors have adequately addressed your comments raised in a previous round of review and you feel that this manuscript is now acceptable for publication, you may indicate that here to bypass the “Comments to the Author” section, enter your conflict of interest statement in the “Confidential to Editor” section, and submit your "Accept" recommendation.

Reviewer #1: All comments have been addressed

Reviewer #2: All comments have been addressed

2. Is the manuscript technically sound, and do the data support the conclusions?

Reviewer #1: Yes

Reviewer #2: Yes

3. Has the statistical analysis been performed appropriately and rigorously? 

Reviewer #1: Yes

Reviewer #2: Yes

4. Have the authors made all data underlying the findings in their manuscript fully available?

Reviewer #1: Yes

Reviewer #2: Yes

5. Is the manuscript presented in an intelligible fashion and written in standard English?

Reviewer #1: Yes

Reviewer #2: Yes

6. Review Comments to the Author

Reviewer #1: Thank you for addressing my concerns. I feel this is a well constructed manuscript that addresses an important research question. Thank you for your contribution.

Reviewer #2: (No Response)

7. PLOS authors have the option to publish the peer review history of their article (what does this mean?). If published, this will include your full peer review and any attached files.

Reviewer #1: No

Reviewer #2: No

---

## [Editor Report · Acceptance letter]

8 Nov 2019

PONE-D-19-14796R1 

Evaluation of forearm vascular resistance during orthostatic stress: velocity is proportional to flow and size doesn’t matter 

Dear Dr. Claydon:

I am pleased to inform you that your manuscript has been deemed suitable for publication in PLOS ONE. Congratulations! Your manuscript is now with our production department. 

With kind regards,

on behalf of

Dr. Christopher Torrens 

Academic Editor

PLOS ONE